

# Quantification of environmentally persistent free radicals and reactive oxygen species in atmospheric aerosol particles

Andrea M. Arangio[1], Haijie Tong[1], Joanna Socorro[1], Ulrich Pöschl[1] & Manabu Shiraiwa[1,*]

[1] Multiphase Chemistry Department, Max Planck Institute for Chemistry, Mainz, Germany

* m.shiraiwa@mpic.de

## Abstract

Fine particulate matter plays a central role in adverse health effects of air pollution. Inhalation and deposition of aerosol particles in the respiratory tract can lead to the release of reactive oxygen species (ROS), which may cause oxidative stress. In this study, we have detected and quantified a wide range of particle-associated radicals using electron paramagnetic resonance (EPR) spectroscopy. Ambient particle samples were collected using a cascade impactor at a semi-urban site in central Europe, Mainz, Germany in May – June 2015. Concentrations of environmentally persistent free radicals (EPFR), most likely semiquinone radicals, were found to be in the range of $(1 - 7) \times 10^{11}$ spins $\mu g^{-1}$ for particles in the accumulation mode, whereas coarse particles with a diameter larger than 1 μm did not contain substantial amounts of EPFR. Using a spin trapping technique followed by deconvolution of EPR spectra, we have also characterized and quantified ROS including OH, superoxide ($O_2^-$) and carbon- and oxygen-centred organic radicals, which were released upon extraction of the particle samples in water. Total ROS amounts of $(0.1 - 3) \times 10^{11}$ spins $\mu g^{-1}$ were released by submicron particle samples and the relative contributions of OH, $O_2^-$, C-centred and O-centred organic radicals were ~11 - 31%, ~2 – 8%, ~41 – 72% and ~0- 25%, respectively, depending on particle sizes. OH was the dominant species for coarse particles. Based on comparisons of the EPR spectra of ambient particulate matter with those of mixtures of organic hydroperoxides, quinones and iron ions followed by chemical analysis using liquid chromatography mass spectrometry (LC-MS), we suggest that the particle-associated ROS were formed by decomposition of organic hydroperoxides interacting with transition metal ions and quinones contained in atmospheric humic-like substances (HULIS).



## 1. Introduction

Epidemiological studies have clearly shown positive correlations between respiratory diseases and ambient fine particulate matter (Pope and Dockery, 2006; Strak et al., 2012; West et al., 2016). A recent study has estimated that outdoor air pollution leads to 3.3 million premature deaths per year worldwide, which is mostly due to particular matter with a particle diameter less than 2.5 μm ($PM_{2.5}$) (Lelieveld et al., 2015). Plausible reasons include the cytotoxicity of ambient $PM_{2.5}$ and its ability to induce inflammatory responses by oxidative stress causing functional alterations of pulmonary epithelial cells (Nel, 2005; Gualtieri et al., 2009). Oxidative stress is mediated by reactive oxygen species (ROS) including OH, $H_2O_2$, superoxide ($O_2^-$), as well as organic radicals (Winterbourn, 2008; Pöschl and Shiraiwa, 2015). Upon PM deposition into the respiratory tract and interactions with lung antioxidants, $H_2O_2$ can be generated by redox-active components contained in $PM_{2.5}$ such as transition metals (Charrier et al., 2014; Fang et al., 2015), semiquinones (Kumagai et al., 1997; Cho et al., 2005; Khachatryan et al., 2011; McWhinney et al., 2013), and humic-like substances (Kumagai et al., 1997; Cho et al., 2005; Lin and Yu, 2011; Charrier et al., 2014; Dou et al., 2015; Fang et al., 2015; Verma et al., 2015b). $H_2O_2$ can be converted into highly-reactive OH radicals via Fenton-like reactions with iron and copper ions (Charrier et al., 2014; Enami et al., 2014).

Ambient particles have been found to contain large amounts of ROS (mostly $H_2O_2$) in the particle phase (Hung and Wang, 2001; Venkatachari et al., 2005; Venkatachari et al., 2007; Fuller et al., 2014). Substantial amounts of particle-bound ROS are found on biogenic secondary organic aerosols (SOA) produced from the oxidation of α-pinene, linalool, and limonene (Chen and Hopke, 2010; Chen et al., 2011; Pavlovic and Hopke, 2011; Wang et al., 2011; Wang et al., 2012). Recently, Tong et al. (2016) have shown that terpene and isoprene SOA can form OH radicals upon interactions with liquid water and iron ions under dark conditions. This can be explained by the decomposition of organic hydroperoxides, which account for the predominant fraction of SOA mass and are generated via multigenerational oxidation and autoxidation (Docherty et al., 2005; Ziemann and Atkinson, 2012; Crounse et al., 2013; Ehn et al., 2014; Epstein et al., 2014; Badali et al., 2015).

In addition, $PM_{2.5}$ contain environmentally persistent free radicals (EPFR) that can be detected directly by electron paramagnetic resonance (EPR) spectroscopy (Dellinger et al., 2001; Khachatryan et al., 2011; Gehling and Dellinger, 2013). EPFR are stable radicals with an e-folding lifetime exceeding one day (Gehling and Dellinger, 2013; Jia et al., 2016). The chemical nature of EPFR is remarkably similar to semiquinone radicals, which can be



stabilized via electron transfer with transition metals in the particle phase (Truong et al., 2010;
Vejerano et al., 2011; Gehling and Dellinger, 2013). EPFR are formed upon combustion and
pyrolysis of organic matter (Dellinger et al., 2001; Dellinger et al., 2007). The formation of
stable radicals can also be induced by heterogeneous and multiphase chemistry of organic
aerosols. Heterogeneous ozonolysis of aerosol particles such as polycyclic aromatic
hydrocarbons (PAH) and pollen proteins can lead to the formation of long-lived reactive
oxygen intermediates (ROI) (Shiraiwa et al., 2011; Shiraiwa et al., 2012; Reinmuth-Selzle et
al., 2014; Borrowman et al., 2015; Kampf et al., 2015; Berkemeier et al., 2016).

In this work, ambient particles with a diameter in the range of 56 nm to 3.2 μm were

collected using a cascade impactor during May – July 2015 in Mainz, Germany. Size
dependences of EPFR concentrations contained in ambient particles have been measured
using an EPR spectrometer. Particles were also extracted in water containing a spin-trapping
agent followed by EPR analysis to quantify the formation of various radical forms of ROS
including OH, superoxide ($O_2^-$) and carbon- and oxygen-centred organic radicals.

## 2. Methods

Ambient particles were collected using a micro-orifice uniform deposition impactor

(MOUDI, 110-R mode, MSP Corporation) on the roof of the Max Planck Institute for
Chemistry, Mainz, Germany (49.99 N, 8.23 E). The sampling was conducted every 24 h
starting at 5 PM during 28 May - 9 June 2015. Particles were collected with a sampling time
of 48 h during 26-27 June and 18-19 July 2015. The sampling was conducted with a flow rate
of 30 L min$^{-1}$ with the following nominal lower cut-off particle diameters: 56, 100, 180, 320,
560 nm, 1 μm, and 1.8 μm. Particles were collected on 47 mm diameter Teflon filters (100
nm pore size, Merck Chemicals GmbH). Each filter was cleaned and sonicated for 10 min
with pure ethanol and ultra-pure water and dried with nitrogen gas before weighing. Teflon
filters were weighed four times using a balance (Mettler Toledo XSE105DU). Particles were
extracted by immersing the filter into a solution containing 350 μL of 20 mM 5-tert-
Butoxycarbonyl-5-methyl-1-pyrroline-N-oxide (BMPO, high purity, Enzo Life Sciences
GmbH) and stirred with a vortex shaker (Heidolph Reax 1) for 7-9 min. BMPO is an efficient
spin-trapping agent for OH, $O_2^-$ and organic radicals (Zhao et al., 2001; Tong et al., 2016).
Extracts were dried for approximately 14-17 min under 1-3 bar $N_2$ flow and the final volume
of the sample for EPR measurements was 50 μL.

A continuous-wave electron paramagnetic resonance (CW-EPR) X-band spectrometer

(EMXplus-10/12, Bruker, Germany) was used for detection and quantification of stable





radicals and ROS. Filters containing particles were folded and introduced into a 4 mm i.d.
quartz tube and inserted directly into a high sensitivity cavity. EPR spectra were recorded at a
room temperature of 23 °C by setting the following operating parameters: a modulation
frequency of 100 kHz; a microwave frequency of 9.84 GHz; a microwave power of 2.149
mW (20 db); a modulation amplitude of 1.0 G; a sweep width of 110.0 G; a sweep time of
175 s; a receiver gain of 40 db; a time constant of 40.96 ms; a conversion time of 160 ms; and
a scan number of 6. The spin-counting method embedded in the Bruker software Xenon was
used to quantify detected radicals. The spin-counting method was calibrated using a standard
compound 4-hydroxy-2,2,6,6-tetramethylpiperidin-1-oxyl (TEMPOL). The detection limit of
EPR was $\sim 1 \times 10^{10}$ spins per μg of particle mass in this study. For better quantification and
determination of the relative contributions of OH, $O_2^-$, carbon-centred and oxygen-centred
organic radicals, EPR spectra were fitted and simulated using Xenon and the Matlab-based
computational package Easyspin (Stoll and Schweiger, 2006).

**3. Results and discussion**
**3.1. Environmentally persistent free radicals**

Figure 1 shows EPR spectra of ambient particles in the lower cut-off diameter range of

56 nm – 1.8 μm. Fine particles, with lower cut-off diameters of 56 - 320 nm, show a single
and unstructured peak with a g-factor of ~2.003 and with a peak to peak distance ($\Delta H_{p-p}$)
ranging from 3 to 8 G. Such spectra are characteristic for EPFR, which have been attributed
to semiquinone radicals (Dellinger et al., 2001; Dellinger et al., 2007; Vejerano et al., 2011;
Bahrle et al., 2015). Particles with a diameter smaller than 56 nm and larger than 560 nm did
not show significant signals, indicating the reduced amount of EPFR in these size ranges.
EPR spectra for particles with the lower cut-off diameters of 56 – 320 nm for each sampling
day are presented in Fig. A1.

The black line in Fig. 2 shows the size distribution of EPFR concentrations. Particles

with different sizes had different radical contents and particles with the lower cut-off
diameter of 100 nm contained the highest EPFR concentrations of $7.0(\pm 0.7) \times 10^{11}$ spins μg$^{-1}$.
High abundances of EPFR in particles in the accumulation mode is consistent with mass size
distributions of combustion-generated particles such as soot or black carbon, which typically
have peak concentrations around 100 – 200 nm (Bond et al., 2013).

Figure 3 shows the temporal evolution of EPFR concentrations contained in particles

with lower cut-off diameters of 100 and 180 nm. During the sampling period of two weeks,
there were two rain events (on 30 May 2015 and 1 June 2015) and three sunny days (4-6 June



2015), and the other days were cloudy. The mass concentrations of particles within the
diameters of 56 - 560 nm were in the range of 3.9 – 12.8 µg m$^{-3}$. Maximum values of ~7 ×
$10^{11}$ spins µg$^{-1}$ were reached during sunny days, indicating that photochemistry may be
related to EPFR production. For example, heterogeneous reactions of photo-oxidants
including $O_3$ and OH with soot or PAH may contribute to the formation of long-lived radicals
(Shiraiwa et al., 2011; Borrowman et al., 2015). Radical concentrations were as low as 6.3 ×
$10^{10}$ spins µg$^{-1}$ during rain events, most likely due to low production of EPFR and scavenging
by precipitation.
EPFR concentrations contained in particles collected for 48 h during 26-27 June 2015
was ~2.2 × $10^{11}$ spins µg$^{-1}$. EPFR concentrations contained in particles within the diameter of
56 – 560 nm averaged over the entire measurement period was 2.0(±1.3) × $10^{11}$ spins µg$^{-1}$.
Squadrito et al. (2001) determined the EPFR concentrations to be in the range of (1-10) ×$10^{11}$
spins µg$^{-1}$ in PM$_{2.5}$ sampled for 24 h in five different urban sites in the United States. Gehling
et al. (2014) reported that the EPFR concentration was in the range of (7-55)×$10^{10}$ spins µg$^{-1}$
at a site in Louisiana near heavy interstate traffic along a major industrial corridor of the
Mississippi River. Shaltout et al. (2015) measured radical concentrations in the range of (2-6)
×$10^{10}$ spins µg$^{-1}$ in PM$_{2.5}$ collected in industrial-, residential- and traffic-dominated sites in
Taif city, Saudi Arabia. These values are comparable with the EPFR concentrations measured
in this work.

**3.2. Reactive oxygen species**
Figure 4a shows EPR spectra of ambient particles with lower cut-off diameters of 56
nm - 1.8 µm extracted in water with the spin-trapping agent BMPO. Each EPR spectrum is
composed of several overlapped lines, originating from different radical forms of ROS.
Dashed lines indicate the positions of each peak for each type of trapped ROS including OH
(green), superoxide (red), carbon-centred (orange) and oxygen-centred organic radicals (blue).
The relative abundance of these radicals was different for each size range, causing the EPR
spectral features to be highly variable. For example, spectra from particles larger than 1.0 µm
consist mainly of four peaks that are typical for OH radicals, whereas those for smaller
particles contain more peaks indicating the presence of multiple radicals.
To estimate the relative amount of each type of ROS, the observed EPR spectra were
fitted and simulated using the software Easyspin 5.0 and Xenon. Four types of radicals have
been used to fit the spectra: BMPO-OH (hyperfine coupling constants of $a^N$ = 14.3 G, $a^H_\beta$ =
12.7 G, $a^H_\gamma$ = 0.61 G), BMPO-OOH ($a^N$ = 14.3 G, $a^H$ = 8.1 G), BMPO-R ($a^N$ = 15.2 G, $a^H$ =





21.6 G) and BMPO-OR ($a^N$ = 14.5 G, $a^H_\beta$ = 16.6 G). As shown in Fig. 4b, the simulated EPR
spectrum reproduced the observed spectrum very well with a small residual. The
deconvolution of spectra allowed us to estimate the relative contribution of four types of ROS
within each particle size range.

Figure 5 shows the relative contributions of OH (green), superoxide (red), carbon-

centred (orange) and oxygen-centred (blue) organic radicals to the total radicals trapped by
BMPO in each particle size range. Carbon-centred radicals are the most abundant type of
radicals, contributing ~50 - 72% of total ROS for PM1. It decreases to 41% and 9% for
particles with lower cut-off diameters of 1 μm and 1.8 μm, respectively. The OH radical
accounts for ~11 – 31% of total trapped radicals for PM1, whereas OH was the dominant
species for coarse particles with diameters of 1.8 – 3.2 μm. The least abundant radical for all
size ranges was $O_2^-$, with contributions of ~2 - 8% and without any clear size dependence.
The amount of oxygen-centred organic radicals ranges between 12% and 25% in particles
with a diameter below 1 μm and its contribution was negligible for coarse particles. Note that
the contribution of oxygen-centred organic radicals for particles with a diameter of 1 – 1.8
μm might be attributed to the OH radical: the hyperfine coupling constants for BMPO-OR for
better fitting the spectrum for this size range needed to be changed slightly ($a^N$ = 13.5 G, $a^H_\beta$
= 15.3 G, $a^H_\gamma$ = 0.6 G). These values are similar to constants of a second conformer of
BMPO-OH.

The red line in Fig. 2 shows the size-dependent concentrations of radical forms of ROS

(e.g., sum of OH, $O_2^-$, C- and O- centred organic radicals). Particles with the lower cut-off
diameter of 100 nm have the highest ROS concentrations of 2.7(±0.2) × $10^{11}$ spins μg$^{-1}$.
Concentrations are smaller for particles in the coarse mode with a diameter larger than 1 μm.
This is consistent with previous studies, suggesting that particles in the accumulation mode
are the most active in ROS generation (Hung and Wang, 2001; Venkatachari et al., 2007;
Saffari et al., 2013; Wang et al., 2013; Saffari et al., 2014). The total concentration of radical
forms of ROS was measured to be 1.2 × $10^{11}$ spins μg$^{-1}$. Note that $O_2^-$ concentrations might
be underestimated as the lifetime of the BMPO-OOH adduct is relatively short (~20 min)
(Ouari et al., 2011; Abbas et al., 2014).

Previous studies have measured redox activity and oxidative potential of PM by the

dichlorofluorescein (DCFH) and dithiothreitol (DTT) assays. The DCFH assay is mostly
sensitive to $H_2O_2$ and other peroxides. For example, Hung and Wang (2001) reported ROS
concentrations as 1×$10^{13}$ μg$^{-1}$ in Taipei, Taiwan. This value is very similar to $H_2O_2$
concentrations contained in ambient $PM_{2.5}$, which has been quantified to be up to 1×$10^{13}$ μg$^{-1}$





in an urban environment in southern California using HPLC fluorescence (Wang et al., 2012).
The DTT assay is based on the decay of DTT due to redox reactions with PM components,
reporting the oxidative potential of PM in moles of DTT consumed per unit of time and mass
of PM. Verma et al. (2015a) and Fang et al. (2015) reported that $PM_{2.5}$ sampled in an urban
environment in Atlanta, Georgia, USA has a DTT activity in the range of 10-70 pmol $min^{-1}$
$\mu g^{-1}$. Assuming an integration time of 20 min needed for the extraction of PM in this work,
this value corresponds to $(1-8) \times 10^{14}$ $\mu g^{-1}$ of DTT molecules consumed. Charrier et al. (2012)
also reported that $PM_{2.5}$ sampled in an urban environment in Fresno, California USA has a
DTT activity of 27 - 61 pmol $min^{-1}\mu g^{-1}$, corresponding to $(2 – 7) \times 10^{14}$ $\mu g^{-1}$ of DTT molecules
consumed in 20 min. These comparisons imply that the total ROS associated with ambient
particles are likely to be dominated by $H_2O_2$, which is about a few orders of magnitude more
abundant than radical forms of ROS determined by EPR in this study. This is reasonable as
$H_2O_2$ is closed shell and much more stable than open-shell radical ROS species.

### 217    3.3. ROS formation mechanism

It has been shown that semiquinones and reduced transition metals including Fe(II) and
Cu(I) can react with $O_2$ to form $O_2^-$, which can be further converted to $H_2O_2$ (Gehling et al.,
2014; Fang et al., 2015). Fenton-like reactions of $H_2O_2$ with Fe(II) or Cu(I) can lead to the
formation of OH radicals (Winterbourn, 2008; Pöschl and Shiraiwa, 2015). OH radicals can
also be generated by the decomposition of organic hydroperoxides (ROOH) contained in
SOA, yielding RO radicals (Tong et al., 2016). Several studies have reported a metal-
independent decomposition of hydroperoxides and organic hydroperoxides driven by
substituted quinones producing RO radicals (Sanchez-Cruz et al., 2014; Huang et al., 2015).
The presence of Fe(II) or quinones is suggested to enhance ROOH decomposition and the
formation of RO and OH radicals (Zhu et al., 2007a; Zhu et al., 2007b; Zhu et al., 2009;
Sanchez-Cruz et al., 2014). Organic peroxides (ROOR) do not yield OH and RO radicals
even in the presence of iron ions (Tong et al., 2016).
Based on these previous studies and considering that ambient particles may contain
quinones, organic hydroperoxides, and transition metals, the observed ROS formation may be
caused by interactions of these chemical components. To further investigate this aspect,
mixtures of organic hydroperoxides, quinones, and Fe(II) were analysed by EPR and liquid
chromatography mass spectrometry (LC-MS). Two standard organic hydroperoxides, cumene
hydroperoxide and tert-butyl hydroperoxide, were used. For quinones, p-benzoquinone and





humic-like substances are used, as HULIS are known to contain substantial amounts of
quinones.
Figure 6 shows the comparison of EPR spectra of ambient particles with a diameter of
180 - 320 nm (black) sampled on 26 June 2015 (same as shown in Fig. 4) and the above
mixtures of organic compounds. Panel (a) includes EPR spectra of mixtures of all three
different components (ROOH, quinone, metal) and panel (b) presents mixtures of two
different components. All three of the organic mixtures in panel (a) resemble the EPR
spectrum of ambient particles by reproducing almost all of the peaks. Particularly, the EPR
spectrum of the mixture containing cumene hydroperoxide, humic acid and Fe(II) closely
overlaps with the ambient particle EPR spectrum. Similarity of spectra between p-
benzoquinone and HULIS suggests that the chemical nature of quinones and HULIS is very
similar. Note that peaks related to the BMPO-OOH adduct at 3497 G and at 3530 G are more
prominent in standard organic mixtures compared to ambient particles. This may be due to
the relatively short lifetime of BMPO-OOH of ~23 min (Zhao et al., 2001), which is
comparable to the extraction and mixing time of BMPO with the atmospheric particles (21 –
28 min), during which BMPO-OOH may have decayed. The trapped radicals have been
further characterized by LC-MS, confirming the presence of OH and semiquinone radicals as
well as carbon- and oxygen centred organic radicals, as detailed in Appendix A and Figs. A1
and A2.
EPR spectra of mixtures containing two compounds in panel (b) reproduce only a part
of the observed peaks. These observations strongly suggest that the combination of these
three chemical components play an important role in generating ROS species by atmospheric
particles. The role of transition metals is crucial to enhance radical formation, most likely via
Fenton-like reactions (Tong et al., 2016) and by participating in redox-cycling of quinones
(Khachatryan and Dellinger, 2011), as intensities of EPR spectra without Fe(II) (CHP +
HULIS, dark blue; CHP + pBq, orange) are small. Carbon-centred radicals may have
multiple sources such as the decomposition of the BMPO-OR adduct by scission of the
carbon in β position, yielding for example $CH_3$ radicals (Zhu et al., 2007b; Huang et al., 2015)
as detected by LC-MS (Fig. A2). They may also be generated by secondary reactions of non-
trapped OH radicals with water-soluble organic compounds.
SOA particles, which may contain large amounts of organic hydroperoxides, account
for a major fraction in PM1 (Jimenez et al., 2009). SOA compounds may also coat coarse
particles such as biological particles (Pöhlker et al., 2012). As shown in Fig. 2, semiquinones
are mostly contained in submicron particles but not in coarse particles. Thus, the release of a



variety of ROS species are most likely due to the interactions of organic hydroperoxides,
semiquinones, and transition metal ions, whereas the dominance of OH radicals in coarse
particles may be due to the decomposition of organic hydroperoxides in the absence of
semiquinones.

**4. Conclusions and implications**

In this study particle-associated environmentally persistent free radicals (EPFR) and

radical forms of ROS have been quantified using electron paramaganetic resonance (EPR)
spectroscopy. EPFR concentrations were measured to be ~$2\times10^{11}$ spins µg$^{-1}$. The chemical
identity of EPFR is likely to be semiquinone radicals based on the g-factors observed by EPR
spectroscopy. We found that particles with different sizes had different radical contents and
particles with a diameter of 100 - 180 nm had the highest abundance of EPFR, whereas
coarse particles did not contain EPFR. This is consistent with the size distribution of
combustion particles such as soot and humic-like substances (HULIS), which may contain
substantial amounts of EPFR.

Reactive oxygen species (ROS) are realeased upon extraction of particles into water.

Particles with the diameter of 100 – 180 nm have released the highest ROS concentrations of
$2.7(\pm 0.2)\times 10^{11}$ spins µg$^{-1}$. By deconvoluting the obtained EPR spectra, four types of radicals,
including OH, O$_2^-$, carbon-centred and oxygen-centred organic radicals were quantified. The
relative amounts of OH, O$_2^-$, C-centred and O-centred organic radicals in submicron particles
were found to be ~11 - 31%, ~2 – 8%, ~41 – 72% and ~0- 25%, respectively, depending on
the particle size. OH was the dominant species for coarse particles with a diameter larger than
1 µm. We suggest that the formation of these ROS species is due to the decomposition of
organic hydroperoxides, which are a major component in SOA, interacting with
semiquinones contained in soot or HULIS. ROS formation can be enhanced in the presence
of iron ions by Fenton-like reactions.

These findings have significant implications for the chemical processing of organic

aerosols in deliquesced particles and cloud water. The released OH radicals within particles
or cloud droplets can oxidize other organic compounds, producing low-volatility products
including organic acids, peroxides, and oligomers (Lim et al., 2010; McNeill et al., 2012;
Ervens, 2015; Herrmann et al., 2015). Autoxidation in the condensed phase might be
triggered by OH radicals forming highly oxidized compounds (Shiraiwa et al., 2014; Tong et
al., 2016). High aqueous oxidant levels may cause fragmentation of organic compounds,
resulting in an increased loss of carbon from the condensed phase (Daumit et al., 2016). The





formed carbon- and oxygen-centred organic radicals are also expected to enhance chemical
aging by participating in particle-phase chemistry involving aldehydes, carbonyls, and
organic peroxides (Ziemann and Atkinson, 2012), although the exact role and impact of
formed organic radicals are still unclear and subject to further studies.
This study suggests that ROS can be generated in lung lining fluid upon inhalation
and respiratory deposition of atmospheric aerosol particles. Even though some fractions of
ROS may be scavenged by antioxidants contained in lung lining fluid, excess concentrations
of ROS including OH radicals, superoxide, and potentially also carbon- and oxygen centred
organic radicals may cause oxidative stress to lung cells and tissues (Winterbourn, 2008;
Pöschl and Shiraiwa, 2015; Tong et al., 2016). ROS play a central role in chemical
transformation of biomolecules such as proteins and lipids in lung fluid to form damage
associated molecular patterns (DAMPs), which can trigger immune reactions causing
inflammation through the toll-like receptor radical cycle (Lucas and Maes, 2013). Due to the
important implications to adverse aerosol health effects, further studies are warranted to
characterize and quantify EPFR and ROS contained in atmospheric aerosol particles in
various locations including highly polluted regions such as in East Asia and India.

## Appendix A. LC-MS analysis of organic mixtures

Two solutions of mixtures of standard organic hydroperoxides and quinones were
analysed by liquid chromatography mass spectrometry (LC-MS). Solution (1) was the
mixture of 200 µL of p-benzoquinone solution at a concentration of 0.2 g L$^{-1}$ (Reagent grade,
≥98%, Sigma-Aldrich) in water (trace SELECT® Ultra, ACS reagent, for ultratrace analysis,
Sigma-Aldrich), 100 µL of Tert-Butyl hydroperoxide solution at a concentration of 8.9 g L$^{-1}$
(Luperox® TBH70X, 70 wt. % H$_2$O, Sigma-Aldrich) in water, 2.5 µL of Iron (II) sulfate
heptahydrate solution at 0.3 g L$^{-1}$ (reagentPlus®, ≥99%, Sigma-Aldrich) in water and 1 mg of
5-tert-Butoxycarbonyl-5-methyl-1-pyrroline-N-oxide (BMPO, high purity, Enzo Life
Sciences GmbH). Solution (2) was the same as solution (1) but without Iron (II) sulfate
heptahydrate. These solutions were stirred with a vortex shaker (Heidolph Reax 1) for 5
minutes.
These solutions were analysed using a 1260 Infinity Bio-inert Quaternary LC system
with a quaternary pump (G5611A), a HiP sampler (G5667A) and an electrospray ionization
(ESI) source interfaced to a Q-TOF mass spectrometer (6540 UHD Accurate-Mass Q-TOF,
Agilent). All modules were controlled by the MassHunter software (B.06.01, Agilent). The
LC column was a Zorbax Extend-C18 Rapid resolution HT (2.1 x 50 mm, 1.8 µm) with a



column temperature of 30 °C. The mobile phases were 3% (v/v) acetonitrile (HPLC Gradient
Grade, Fisher Chemical) in water with formic acid (0.1 % v/v, LC-MS Chromasolv, Sigma-
Aldrich) (Eluent A) and 3 % water in acetonitrile (Eluent B). The injection volume was 10
µL. The flow rate was 0.2 mL min$^{-1}$ with a gradient program that starting with 3 % B for 3
min followed by a 36 minutes step that raised eluent B to 60 %. Further, Eluent B was
increased to 80 % at 40 minutes and returned to initial conditions within 0.1 minutes,
followed by column re-equilibration for 9.9 min before the next run.
The ESI-Q-TOF instrument was operated in the positive ionization mode (ESI+) with a
gas temperature of 325 °C, 20 psig nebulizer, 4000 V capillary voltage and 90 V fragmentor
voltage. During the full spectrum MS mode, no collision energy was used in order to collect
species as their molecular ions. During MS/MS analysis employed for the structure
determination, the fragmentation of protonated ions was conducted using the target MS/MS
mode with 20 V collision energy. Spectra were recorded over the mass range of $m/z$ 50-1000.
Data analysis was performed using qualitative data analysis software (B.06.00, Agilent).
Blank solutions without BMPO were also prepared and analysed. Background signals were
subtracted from the MS spectrum.
Figure A2 shows LC-MS/MS mass spectra of the products formed from the reaction of
tert-butyl hydroperoxide, p-benzoquinone, BMPO in the presence of iron (solution 1). Very
similar results were obtained for solutions in the absence of iron (solution 2). BMPO adducts
with radicals ·OH, ·CH$_3$ and ·OCH$_3$ were identified by LC-MS/MS. As shown in Figure
A2(a.1), it was observed that ions at $m/z$ 160.0596, 216.1221 and 238.1020 were majors ions
formed in the positive mode. These protonated ions represent the [BMPO+OH-C$_4$H$_8$+H]$^+$,
[BMPO+OH+H]$^+$ and [BMPO+OH+Na+H]$^+$ spin adducts, respectively. Figure A2 (a.2)
displays the mass spectrum in the MS/MS mode for the fragmentation of the ion $m/z$
216.1221. Results confirmed the loss of the t-butoxycarbonyl function (- C$_4$H$_8$), which is a
characteristic fragment of BMPO, to form the ion $m/z$ 160.0585. The observed ion fragment
$m/z$ 114.0544, can be formed by the loss of CH$_2$O$_2$, as shown in Fig. A2(a.3). In Fig. A2(b.1),
the spectrum showed the mass $m/z$ 158.0804 and 214.1431 that can be attributed to the
[BMPO+CH$_3$-C$_4$H$_8$+H]$^+$ and [BMPO+CH$_3$+H]$^+$, respectively. The most abundant fragment
ion ($m/z$ 158.0803) in the MS/MS mode confirmed the formation of BMPO+CH$_3$ adduct, as
shown in Fig. A2(b.2). The peak $m/z$ 112.0752 can be formed by the loss of CH$_2$O$_2$ (Fig.
A2(b.3)). The spectrum in Fig. A2(c.1) shows major peaks at $m/z$ 174.0752, 230.1378 and
252.1198, corresponding to [BMPO+OCH$_3$-C$_4$H$_8$+H]$^+$, [BMPO+OCH$_3$+H]$^+$ and
[BMPO+OCH$_3$+Na+H]$^+$, respectively. The formation of BMPO-OCH$_3$ was confirmed in





MS/MS by the loss of the t-butoxycarbonyl functional group of BMPO to form the ion at $m/z$
174.0749 (panels (c.2) and (c.3)).

In addition, the radicals $C_6H_5O_2\cdot$ or $\cdot C_6H_5O_2$ and $C_6H_9O_2\cdot$ or $\cdot C_6H_9O_2$ were detected,

although it was not possible to determine whether the chemical structure represented carbon-
or oxygen-centred organic radicals using the applied method. Figure A3(a.1) shows the
formation of protonated ions $[BMPO+C_6H_5O_2+H]^+$ and $[BMPO+C_6H_5O_2+Na+H]^+$ with $m/z$
308.1475 and 330.1298, respectively. The fragmentation in the MS/MS mode confirms the
formation of $BMPO+C_6H_5O_2$ ($m/z$ 252.0855) that correspond to the loss of the characteristic
t-butoxycarbonyl function as show in Fig. A3(a.2).  The ion fragment observed m/z 128.0702,
can be formed by the loss of $C_5O_4$ (Figure A3(a.3)). Figure A3(b.1) shows the ion m/z
312.1789, which can be attributed to the $BMPO+C_6H_9O_2$ spin adduct. Figure A3(b.2)
suggests that the fragmentation of $m/z$ 312.1789 to $m/z$ 256.1166 by the loss of - $C_4H_8$ (Figure
A3(b.3)).

**Acknowledgements.**
This study is funded by the Max Planck Society. We thank Christopher Kampf and Fobang
Liu for helping LC-MS analysis and Pascale Lakey for helpful comments.

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





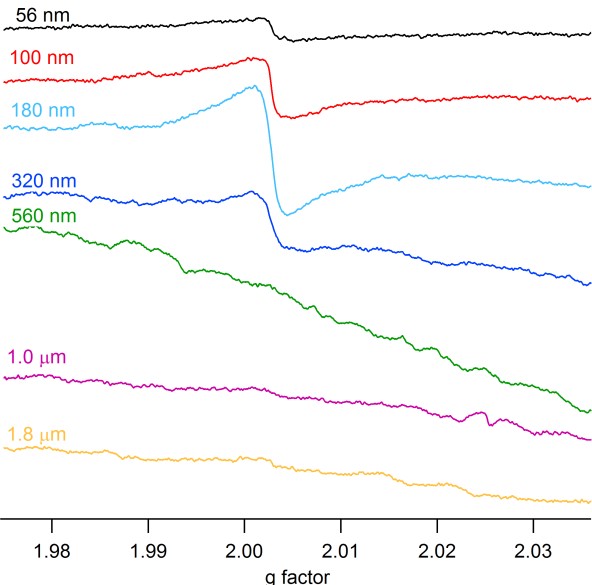


**Figure 1**: Electron paramagnetic resonance (EPR) spectra of atmospheric aerosol impactor

samples with lower cut-off diameters in the range of 56 nm to 1.8 micrometer collected in

Mainz, Germany during 26 - 27 June, 2015.



**Figure 2**: Concentrations (spins per microgram of particles) of environmentally persistent

free radicals (EPFR) and radical forms of reactive oxygen species (ROS) in atmospheric

aerosol samples plotted against particle diameter.




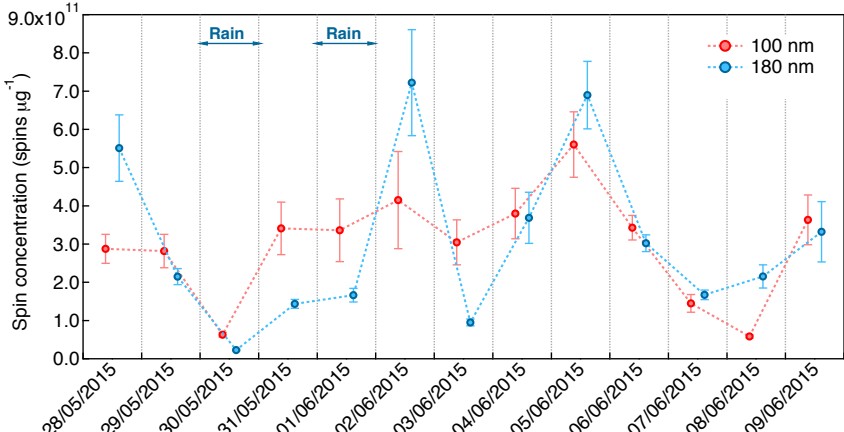


**Figure 3:** Temporal evolution of concentrations of environmentally persistent free radicals

(EPFR) contained in atmospheric aerosol samples with lower cutoff diameters of 100 nm (red)

and 180 nm (blue), measured in Mainz, Germany during May – June 2015.


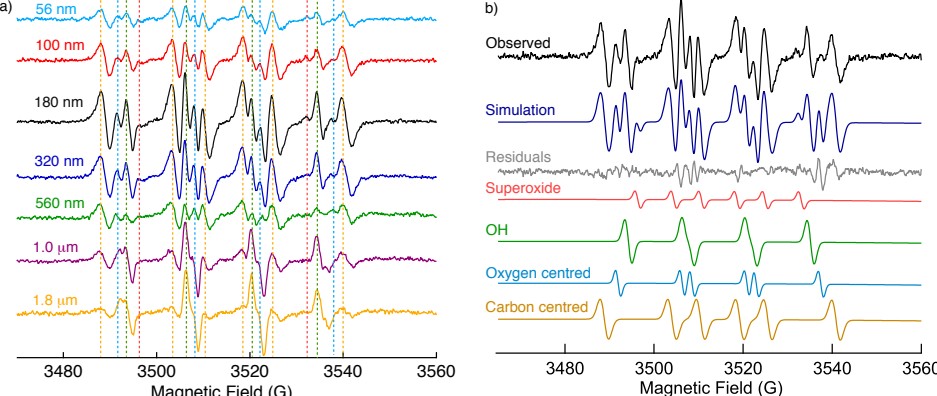


 **Figure 4**: a) Electron paramagnetic resonance (EPR) spectra of ambient aerosol impactor

samples (Mainz, Germany, 26-27 June 2015) with lower cut-off diameters in the range of 56

nm to 1.8 μm extracted in water mixed with the spin-trapping agent BMPO. Dashed lines

indicate the position of each peak for different types of trapped radicals of $O_2^-$ (red),  OH

(green), carbon-centred (orange), and oxygen-centred organic radicals (light blue). b)

Simulation of the EPR spectrum of the atmospheric aerosol impactor sample with particle

diameters in the range of 180-320 nm (lower to upper cut-off) by deconvoluting into $O_2^-$, OH,

O-centred and C-centred organic radicals (Blue = synthesis, grey = residual).

636



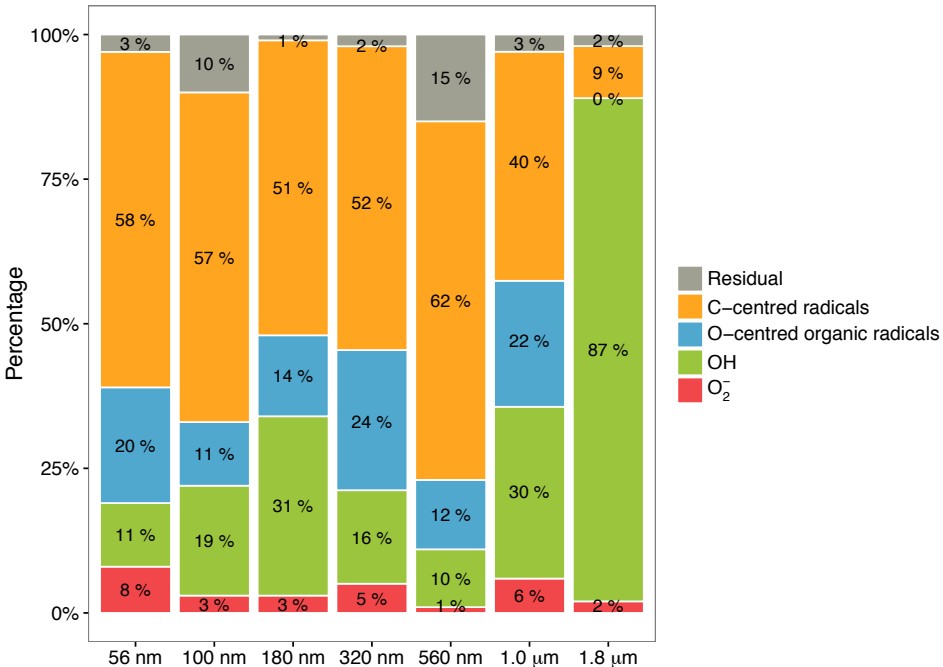

637

**Figure 5:** Relative amount of ROS in atmospheric aerosol impactor samples with lower cut-
off diameters in the range of 50 nm -1.8 µm (Mainz, Germany, 26-27 June 2015): $O_2^-$ (red),
OH (green), carbon-centred (orange), oxygen-centred organic radicals (blue) and redisual
(unidentified, grey).

642



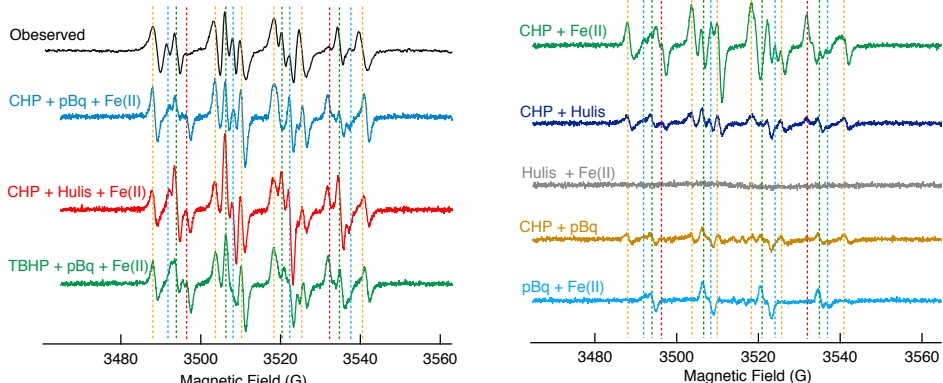

643

**Figure 6:** Electron paramagnetic resonance (EPR) spectra of atmospheric aerosol impactor sample with particle diameters in the range of 180-320 nm (lower to upper cut-off) extracted with water and BMPO (black) and of aqueous substance mixtures with the following ingredients: cumene hydroperoxide (CHP), p-benzoquinone (pBq) and Fe(II) (light blue), t-butyl hydroperoxide (TBHP). The dashed vertical lines indicate the main peaks of BMPO adducts with $O_2^-$ (red), OH (green), carbon- (light blown), and oxygen-centred organic radicals (light blue).










**Figure A1:** EPR spectra of atmospheric aerosol impactor samples with lower cut-off diameters of 56 nm (black), 100 nm (red), 180 nm (light blue) and 320 nm (green) for the measurement period during 28 May – 9 June 2015.



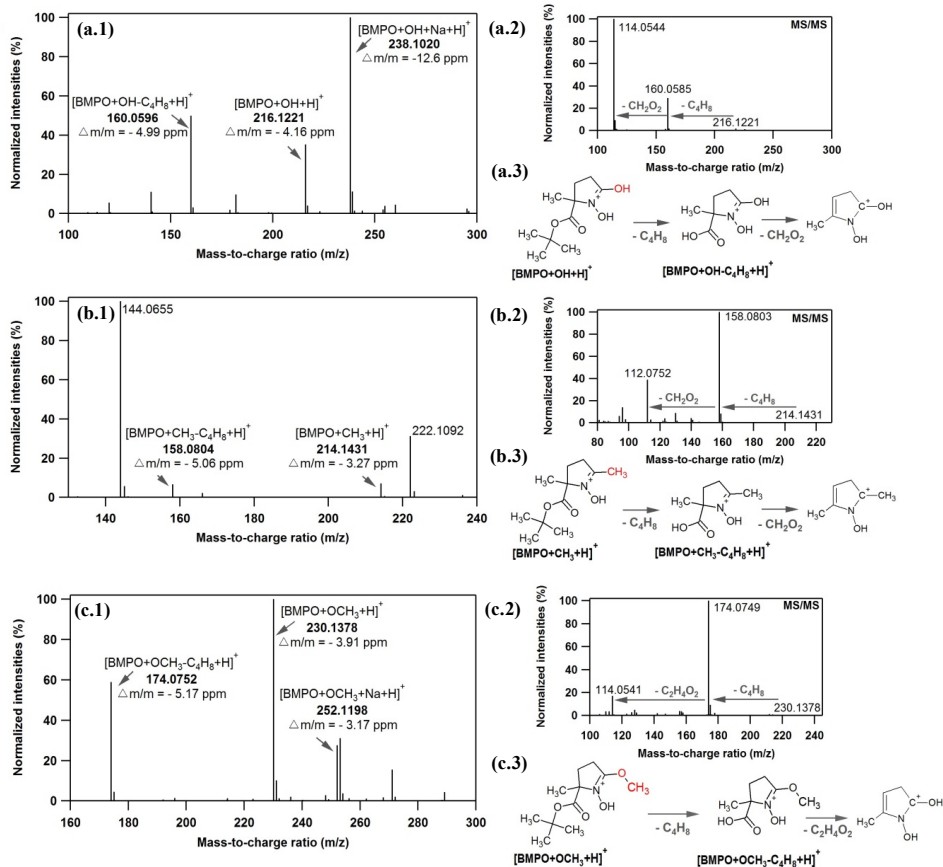

**Figure A2:** Mass spectra obtained with LC-MS/MS in the positive ionization mode from the mixture of tert-butyl hydroperoxide, p-benzoquinone and BMPO in the presence of iron (solution (1)). MS spectra of **(a.1)** BMPO+OH, **(b.1)** BMPO+CH₃ and **(c.1)** BMPO+OCH₃. MS/MS spectra of **(a.2)** BMPO+OH, **(b.2)** BMPO+CH₃, and **(c.2)** BMPO+OCH₃. Proposed fragmentation pathways of **(a.3)** BMPO+OH, **(b.3)** BMPO+CH₃ and **(c.3)** BMPO+OCH₃.





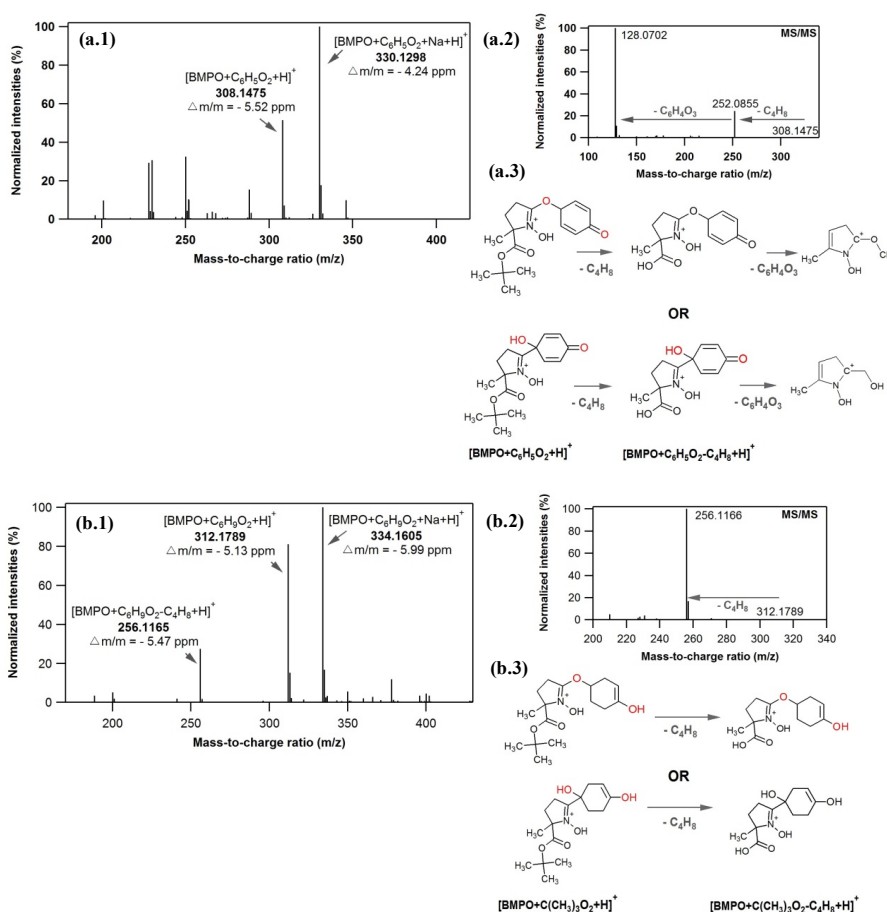

667

**Figure A3:** Mass spectra obtained with LC-MS/MS in the positive ionization mode for
solution (1). MS spectra of **(a.1)** BMPO+$C_6H_5O_2$ and **(b.1)** BMPO+$C_6H_9O_2$. MS/MS spectra
of **(a.2)** BMPO+$C_6H_5O_2$ and **(b.2)** BMPO+$C_6H_9O_2$. Proposed fragmentation pathway of **(a.3)**
BMPO+$C_6H_5O_2$ and **(b.3)** BMPO+$C_6H_9O_2$.

672