# Peer review of "Quantification of environmentally persistent free radicals and reactive oxygen species in atmospheric aerosol particles"

_Atmospheric Chemistry and Physics, 2016_

## Referee Comment (RC1) · Anonymous Referee #1 · 8 Sep 2016

The paper from Arangio et al. measures the concentration of environmentally persistent free radicals (EPFR) and reactive oxygen species (ROS) in size segregated ambient aerosols. EPFR were measured directly by EPR spectrometer, while the ROS were measured by extracting the particles in water and then EPR analysis. As per the reviewer's knowledge, this is first comprehensive measurement of EPFR and ROS in size-segregated aerosols. ROS are an important species in ambient aerosols and could be biologically relevant. In addition to these novel measurements, authors also throw lights on the possible mechanisms of ROS generation through redox cycling between organic compounds and transition metals. An improved understanding of these mechanisms is important to comprehend the aging process of atmospheric aerosols.

[Figure]

The paper is well written and easily comprehensible. Therefore, I recommend the publication of this manuscript. However, I have few comments which the authors should consider to make their work better:

Page 2, Line 42: Are there literature evidences that organic radicals also mediate in the oxidative stress? If yes, then authors should include them.

Page 3, Line 87: Why these two samples were collected for a longer duration? Are the authors not concerned about the loss of semivolatiles during that long sampling duration?

Page 4, Line 128: Authors should somewhere explain these units of spins $\mu$g-1, probably in the method section.

Page 4, Line 129-131: Can authors elaborate on their sentence that EPFR distribution is similar to soot? Do you mean that there is commonality in the sources of two?

Page 5, Line 143: Why the samples collected on these two days are significant and discussed separately?

Page 5, Line 143-150: I am not sure why the authors have discussed the sampling duration separately. The EPFR concentration expressed in units of spin/$\mu$g should not be affected by the sampling duration.

Page 6, Line 176: Is it 41

Page 6, Line 201: What are the units here for ROS, is it spins/$\mu$g?

Page 7, Line 203-215: I think the authors are completely confused here. DTT assay doesn't measure the ROS in the particle, rather the capability of particles to generate ROS in surrogate biological environment. I am not clear what the authors want to deduce in this discussion and what is the significance of this number of (2-7) x 1014 ug-1 of DTT molecules? It is important to note that DTT activity is a completely arbitrary unit and depends on the initial DTT concentration used in the assay. Page 8, Line 237:

Can the authors add references showing HULIS is known to contain substantial amount of quinones?

Page 10, Line 308-310: I don't think that this study shows that ROS can be generated in lung fluid. I think again the authors are confusing between ROS activity (capability of particles to generate ROS) vs. ROS on the particles (measured in this study).

Please also note the supplement to this comment:
http://www.atmos-chem-phys-discuss.net/acp-2016-501/acp-2016-501-RC1-supplement.pdf

---

## Referee Comment (RC2) · Anonymous Referee #2 · 9 Sep 2016

This study reports concentrations of particle-bound environmentally persistent free radicals (EPFR) and radical forms of reactive oxygen species (ROS) using electron paramagnetic resonance (EPR) spectroscopy. ROS species quantified after release by extraction of submicron particle samples in water include OH, O2-, carbon- and oxygen-centered organic radicals; the authors further report concentrations as a function of particle size. The study proposes that the formation of ROS is due to the decomposition of organic hydroperoxides interacting with semiquinones in soot and/or HULIS particles, while EPFR are likely from semiquinone radicals. The study is well written and relevant to the atmosphere and human health concerns. I recommend publication in ACP after the following questions and comments are addressed.

Comments:

-How do the concentrations of ROS (spins $\mu$g-1) compare to those previously reported? It may be useful to include a note in the Methods section about the context of these units in terms of their relationship to standard particle concentrations.

-A mention of g-factor before the Results section may be helpful—as written it is difficult to understand the importance of the parameter and how unique a g-factor is to each measurable species.

-Figure 1 - what does the structure at 560 nm indicate?

-Lines 83-98. Can you expand a bit on any transmission effects of the impactor, especially for the coarse particles?

-Figure 2. What do the error bars indicate? Is there any significance that both ROS and EPFR have minima at the same size (560 nm)?

-Lines 99-113. What are the background concentrations of these species? Is there any signal when EPFR are not present?

-Figure 3. Can the authors expand on why rain events do not seem to dampen concentrations of EPFR in 100 nm particles as much as 180 nm particles?

-Figure 5. The authors may find it useful to note the total ROS concentrations to further illustrate the size dependence. For the largest particles (1.8 $\mu$m especially), OH seemingly dominates the total ROS concentrations - did OH significantly contribute to the total ROS concentration at 1.8 $\mu$m or is this due to the smaller ROS concentrations skewing the total contributions of each species?

-Lines 308-319. Is our lung capacity inhalation dependent on the total concentration (spins $\mu$g-1) of these ROS/EPFR species? Is there an amount of ROS/EPFR that our lungs can safely inhale without potential health harm?

---

## Referee Comment (RC3) · Anonymous Referee #3 · 11 Sep 2016

In this study, the authors report size-resolved measurements of EPFR and ROS radicals using EPR spectroscopy and LC-MS for samples collected for one week in Mainz as a demonstration. They show size-dependent variations in the proportion of radicals of ROS and EPFR, with concentrations of both peaking in the accumulation mode, and find that carbon-centered radicals contribute the largest proportions to radical species in ROS for PM1. Using laboratory generated spectra, they further propose that mechanisms for ROS generation in these samples require a combination of transition metals with organic hydropreoxides and quinones. The work is of high technical quality with important implications for understanding atmospheric processes and air quality, and the manuscript is well-written. The work is thus recommended for publication in Atmo-

spheric Chemistry and Physics after the following comments have been addressed.

General comments regarding the measurement method:

1. Are there any transformation artifacts from the initial measurement of EPFR? For example, if the authors use collocated measurements and analyze ROS directly on the second filter, would they expect to find the same ROS concentrations measured after the filter has been used for quantification of EPFR?

2. On p.3, line 95, it is stated that "BMPO is an efficient spin-trapping agent [...]." Is the efficiency effectively considered to be 100\% for all, or are there biases for certain radicals?

3. In the drying process with N2, is it possible that negative artifacts are introduced? How is the dried extract introduced into the EPR spectrometer?

4. Is the detection limit reported the instrument detection limit, or analytical detection limit derived from blanks?

General comments regarding the reported concentrations:

1. Would it be meaningful to plot radiation intensity alongside Fig. 3 to discuss the potential role of photochemistry? For instance, on 02/06/2015, the concentration is also high even though the conditions are presumably cloudy according to descriptions in text. In this regard, the radical concentrations appear to depend on many factors and underscores the benefit of integrating this technique into larger measurement campaigns.

2. Are the proportions in Fig. 5 meant to be representative of those observed during the entire measurement campaign?

Minor comments:

1. p. 3, line 92: The authors discuss pre-cleaning and weighing, and then discuss particle extraction. It would be helpful if the description were explicit in the pre-sampling

and post-sampling procedures.

2. p. 5, line 152: This is just a semantic issue, but it would seem more appropriate to say that the values in this work are comparable with the EPFR concentrations measured by Shaltout et al. (2015) - instead of the other way around - since their work preceded this one and sets the precedent to which following studies should be compared.

3. In the conclusions, the measurement location and period should be restated so the reported concentrations are placed in the proper context.
* * *

---

## Referee Comment (RC4) · Anonymous Referee #4 · 11 Sep 2016

This paper describes measurements of the concentrations of reactive species embedded in atmospheric aerosol collected from the roof of the MPI. It is a followup to the earlier paper this year by Tong et al which compared field samples to lab samples collected for a shorter period of time and focused on OH generation. The measurement method consists of extracting soluble molecules from the particles and reacting them with a scavenger. This is essentially a physical chemistry paper and my comments are from that perspective. This paper will be publishable after some edits to respond to the following comments.

(1) I found the terminology used by the authors to be confusing in places. Through use of words such as "we have also characterized and quantified ROS including OH,

superoxide (O2-) and carbon- and oxygen-centered organic radicals, which were re-leased upon extraction of the particle samples in water." the reader could conclude that the radicals are persistently present in the particle, rather than being formed by reaction of water with precursors during the extraction process and later scavenged. The multiple chemical steps involved in the experiment lend an ambiguity to how to relate the lab processes that are responsible for formation of detectable spins to atmo-spheric and physiological processes. As another example, on pages 4-5 the authors talk about spins per microgram but do not define this quantity. I presume they mean spins detected by extraction from a sample of this mass using the protocol described. The concentration of the scavenger is not given so it is not clear how closely this pro-cess is controlled or how repeatable it is. Clarification of the terminology and relation of the experimental conditions to those found in the lung and in clouds, for example, would be helpful.

(2) Only particles smaller than 1 micron contain extractable ROS material. Do the authors understand why this is? Since peroxides are photo labile I might have expected the opposite - the larger, more optically opaque particles would have more precursors than the smaller ones assuming the extraction processes work the same way for all particle sizes. Some discussion of the size effects would be useful.

(3) The reactive oxygen species released and scavenged during the analytical protocol are well known to have rich chemistry in water and very different reactivities compared to each other. Have the authors determined how efficiently are they being detected (absolute and relative values)?

(4) On page 9 line 269ff there is a section discussing implications for aerosol chemistry and lung chemistry. Since there are no data in this paper specifically looking at these implications but there are in Tong I recommend the reader referred back to the earlier paper instead.
* * *

---

## Author Comment (AC1) · 4 Oct 2016

**Response to the comment to the Anonymous Referee #1**

The paper from Arangio et al. measures the concentration of environmentally persistent free radicals (EPFR) and reactive oxygen species (ROS) in size segregated ambient aerosols. EPFR were measured directly by EPR spectrometer, while the ROS were measured by extracting the particles in water and then EPR analysis. As per the reviewer's knowledge, this is first comprehensive measurement of EPFR and ROS in size-segregated aerosols. ROS are an important species in ambient aerosols and could be biologically relevant. In addition to these novel measurements, authors also throw lights on the possible mechanisms of ROS generation through redox cycling between organic compounds and transition metals. An improved understanding of these mechanisms is important to comprehend the aging process of atmospheric aerosols. The paper is well written and easily comprehensible. Therefore, I recommend the publication of this manuscript. However, I have few comments which the authors should consider to make their work better:

Response:
We thank the referee's review and very positive evaluation of this manuscript. The point-by point responses are given below.

Page 2, Line 42: Are there literature evidences that organic radicals also mediate in the oxidative stress? If yes, then authors should include them.

Response:
Some types of organic radicals such as semiquinone and phenoxy radicals are known to play a role in oxidative stress (Pryor et al., 1995; Winterbourn, 2008; Birben et al., 2012). We have added new references.

Page 3, Line 87: Why these two samples were collected for a longer duration? Are the authors not concerned about the loss of semivolatiles during that long sampling duration?

Response:
We agree that semi-volatile compounds may be lost for long sampling duration, which is a common problem of the particle collection using an impactor. Two samples were collected for 48 h in order to obtain sufficiently high mass loadings for all particle size ranges. We have clarified this point in the revised manuscript.

Page 4, Line 128: Authors should somewhere explain these units of spins μg-1, probably in the method section.

Response:
The unit spins $\mu g^{-1}$ indicates the number of spins (or radicals) per μg of particle mass. We will clarify it in the revised manuscript.

Page 4, Line 129-131: Can authors elaborate on their sentence that EPFR distribution is similar to soot? Do you mean that there is commonality in the sources of two?

Response:
Yes, we think that the sources of soot and EPFR are very similar (e.g., combustion) and EPFR may be often associated with soot particles (Dellinger et al., 2001). We will clarify this point in the revised manuscript.

Page 5, Line 143: Why the samples collected on these two days are significant and discussed separately?

Response:
As explained above, for certain periods we have collected particles for 48 h to collect enough particle mass to perform EPFR and ROS analysis for wide particle diameters (50 nm - 1.8 μm). On the other hand, particles collected for 13 days with a sampling time of 24h were focused on limited particle size range of 50 nm to 500 nm diameter particles.

Page 5, Line 143-150: I am not sure why the authors have discussed the sampling duration separately. The EPFR concentration expressed in units of spin/μg should not be affected by the sampling duration.

Response:
Indeed the EPFR concentrations are not affected by the sampling duration, but the particle diameters were different. We will clarify it by including the below sentences in the revised manuscript:

"EPFR concentrations contained in particles within the diameter of 50 nm – 3.2 μm collected for 48 h during 26-27 June 2015 was ~$2.2 \times 10^{11}$ spins $\mu g^{-1}$. EPFR concentrations contained in particles

within the diameter of 56 – 560 nm averaged over the entire measurement period was 2.0(±1.3) × $10^{11}$ spins μg$^{-1}$."

Page 6, Line 176: Is it 41

Response:
Carbon-centred radicals are reduced to 40% in the 1 μm stage. Thanks for point out this typo, we will correct it in the revised manuscript.

Page 6, Line 201: What are the units here for ROS, is it spins/μg?

Response:
The unit is μg$^{-1}$ and not spins μg$^{-1}$, as $H_2O_2$ is not radical, but it can be still directly compared with concentrations of EPFR and radical forms of ROS (in the unit of spins μg$^{-1}$) as measured in this study.

Page 7, Line 203-215: I think the authors are completely confused here. DTT assay doesn't measure the ROS in the particle, rather the capability of particles to generate ROS in surrogate biological environment. I am not clear what the authors want to deduce in this discussion and what is the significance of this number of (2-7) x 10^14 ug-1 of DTT molecules? It is important to note that DTT activity is a completely arbitrary unit and depends on the initial DTT concentration used in the assay.

Response:
The DTT assay measures the consumption rate of DTT molecules due to reactions of redox-active components of particulate matter with antioxidants. The total number of DTT molecules consumed per unit of mass and time are measure of the redox activity or oxidative potential of chemical compounds contained in the particles. The underlying assumption of the DTT assay is that the consumption of one DTT molecule would lead to the generation of one ROS molecule (e.g., $H_2O_2$). We agree that this assumption has not proved robustly and we are actually planning to investigate this aspect in details in the follow-up study. We think it is still meaningful to make this comparison, but we will refine the sentence to avoid confusion in the revised manuscript.

Page 8, Line 237:Can the authors add references showing HULIS is known to contain substantial amount of quinones?

Response:

We have added the following reference:

Verma, V., Wang, Y., El-Afifi, R., Fang, T., Rowland, J., Russell, A. G., and Weber, R. J.: Fractionating ambient humic-like substances (HULIS) for their reactive oxygen species activity – Assessing the importance of quinones and atmospheric aging, Atmos. Environ., 120, 351-359, 2015.

Page 10, Line 308-310: I don't think that this study shows that ROS can be generated in lung fluid. I think again the authors are confusing between ROS activity (capability of particles to generate ROS) vs. ROS on the particles (measured in this study).

Response:

As pointed out, this study itself did not show that ROS can be generated in the lung lining fluid containing antioxidants, but it did show that the particles can form ROS in water. Several previous studies have shown that redox-active components such as transition metals and quinones can induce formation of ROS species upon interactions with lung antioxidants (Charrier et al., 2014; Charrier and Anastasio, 2011). We will clarify it in the revised manuscript as below:

"Previous studies have shown that redox-active components such as transition metals and quinones can induce ROS formation in surrogate lung lining fluid upon interactions with antioxidants (Charrier et al., 2014; Charrier and Anastasio, 2011). This study also implies that ROS may be released in lung lining fluid upon inhalation and respiratory deposition of atmospheric aerosol particles."

**References:**

Charrier, J. G., and Anastasio, C.: Impacts of antioxidants on hydroxyl radical production from individual and mixed transition metals in a surrogate lung fluid, Atmos. Environ., 45, 7555-7562, 2011.

Charrier, J. G., McFall, A. S., Richards-Henderson, N. K., and Anastasio, C.: Hydrogen Peroxide Formation in a Surrogate Lung Fluid by Transition Metals and Quinones Present in Particulate Matter, Environ. Sci. Technol., 48, 7010-7017, 2014.

Birben, E., Sahiner, U. M., Sackesen, C., Erzurum, S., and Kalayci, O.: Oxidative stress and antioxidant defense, World Allergy Organization Journal, 5, 1, 2012.

Pryor, W. A., Squadrito, G. L., and Friedman, M.: The cascade mechanism to explain ozone toxicity - The role of lipid ozonation products, Free Radical Biol. Med., 19, 935-941, 1995.

Winterbourn, C. C.: Reconciling the chemistry and biology of reactive oxygen species, Nature Chem. Biol., 4, 278-286, 2008.

---

## Author Comment (AC2) · 4 Oct 2016

**Response to the comment to the Anonymous Referee #2**

This study reports concentrations of particle-bound environmentally persistent free radicals (EPFR) and radical forms of reactive oxygen species (ROS) using electron paramagnetic resonance (EPR) spectroscopy. ROS species quantified after release by extraction of submicron particle samples in water include OH, O2-, carbon- and oxygen-centered organic radicals; the authors further report concentrations as a function of particle size. The study proposes that the formation of ROS is due to the decomposition of organic hydroperoxides interacting with semiquinones in soot and/or HULIS particles, while EPFR are likely from semiquinone radicals. The study is well written and relevant to the atmosphere and human health concerns. I recommend publication in ACP after the following questions and comments are addressed.

Response:
We thank the referee for review and very positive evaluation of this manuscript.

Comments:

-How do the concentrations of ROS (spins µg-1) compare to those previously reported? It may be useful to include a note in the Methods section about the context of these units in terms of their relationship to standard particle concentrations.

Response:
This study provided the concentrations of radical forms of ROS (e.g., sum of OH, $O_2^-$, C- and O-centered organic radicals), which is the first measurement to the best of our knowledge. Thus, we cannot make direct comparison with previous studies. Instead, we have made comparison with previous measurements of redox activity and oxidative potential of PM by the dichlorofluorescein (DCFH) and dithiothreitol (DTT) assays, as discussed in Sect. 3.2. Assuming that the consumption of one DTT molecule would correspond to the generation of one ROS molecule (e.g., $H_2O_2$), these values are about a few orders of magnitude higher than concentrations of radical forms of ROS measured in this study. This is reasonable as $H_2O_2$ is closed shell and much more stable than open-shell radical ROS. The standard unit for the particle mass concentration is µg m$^{-3}$, which indicates µg of particle mass in 1 m$^3$ of air; whereas the unit (spins µg$^{-1}$) of ROS or EPFR concentrations used in this study indicates the number of spins (or radicals) per unit of particle mass. We will clarify this point in the method section of the revised manuscript.

-A mention of g-factor before the Results section may be helpful as written it is difficult to understand the importance of the parameter and how unique a g-factor is to each measurable species.

Response:

Thanks for your suggestion. We will add the following sentence in the method section:

"Paramagnetic species are characterized based on their g-factor values. Free electrons have a g-factor value of 2.0023 and organic radicals have higher g-factor values (2.0030 – 2.0060), depending on the number of oxygen atom in the molecule (B. Dellinger et al. 2007)"

-Figure 1 - what does the structure at 560 nm indicate?

Response:

The spectrum for 560 nm particles indicates that no radicals have been found for this size range particles. The high background, probably due to metal oxides, causes the spectrum to be steep in shape and the fine structure seems to be not significant.

-Lines 83-98. Can you expand a bit on any transmission effects of the impactor, especially for the coarse particles?

Response:

For impactor such as MOUDI, the transmission effect of particles collected on a stage is considered to be a step function with boundaries of the steps corresponding to the cut-off sizes of the upper and lower stages. That is, each stage has an efficiency of 50% in collecting particles with the diameter comparable to its nominal cut-off size. The collection efficiency reaches 100% for particles with diameter just bigger than the nominal cut-off size of the upper stage (Marple et. al. 1991). However, the collection efficiency of particles with the diameter comparable to the nominal cut-off size of a stage can be lower than 50% due to bouncing effects. Due to bouncing, particles that are supposed to impact on a stage can be transferred to the next stage. This process is much more influential for particles in the coarse fraction. We will add the below sentence in the revised manuscript:

"Note that transmission and bouncing effects may cause mixing of particles exhibiting relatively different sizes on one stage, particularly for coarse particles (Gomes et al., 1990; Bateman et al., 2014)."

-Figure 2. What do the error bars indicate? Is there any significance that both ROS and EPFR have minima at the same size (560 nm)?

Response:

The error bars represent standard errors based on uncertainties of the particle mass and signal integration of EPR spectra. We will add this information in the figure caption. The minimum in the EPFR and ROS concentration for particles at 560 nm is likely due to a low mass loading in this stage.

-Lines 99-113. What are the background concentrations of these species? Is there any signal when EPFR are not present?

Response:

Blank measurements confirmed that there are no background concentrations of EPFR. In the absence of EPFR, the signal is just a horizontal line, when the concentrations of other paramagnetic species (e.g., transition metals) are below the detection limit.

-Figure 3. Can the authors expand on why rain events do not seem to dampen concentrations of EPFR in 100 nm particles as much as 180 nm particles?

Response:

EPFR concentrations in both 100 and 180 nm particle decreased substantially after rain events on May 30 (Saturday), but it was not very obvious, on June 1 (Monday), 2015. EPFR concentrations are controlled by both emission and deposition. The road traffic can be a main contributor of $PM_{2.5}$ in the area around the sampling site and it is generally very limited during the weekend. This means that emission or production rates of EPFR on June 1 must be higher than on May 30. Moreover, scavenging efficiency of particles depends on rainfall intensity and sizes of rain droplet and particle (Seinfeld & Pandis, 2006), which might have caused the difference for 100 and 180 nm particles.

Figure 5. The authors may find it useful to note the total ROS concentrations to further illustrate the size dependence. For the largest particles (1.8 μm especially), OH seemingly dominates the total ROS concentrations

Response:

The total ROS concentrations as a function of the particle diameter are shown as the red line in Figure 2. Figure 5 shows the relative contribution of each type of ROS to the total amount of ROS at each stage.

-did OH significantly contribute to the total ROS concentration at 1.8 μm or is this due to the smaller ROS concentrations skewing the total contributions of each species?

Response:
Yes, OH contributed up to about 90 % of the total ROS released by 1.8 μm particles.

-Lines 308-319. Is our lung capacity inhalation dependent on the total concentration (spins μg-1) of these ROS/EPFR species? Is there an amount of ROS/EPFR that our lungs can safely inhale without potential health harm?

Response:
This is a very interesting and important question to be addressed. We have recently reported that fine particulate matter (PM2.5) containing redox-active transition metals, quinones, and secondary organic aerosols can increase ROS concentrations in the lung lining fluid to levels characteristic for respiratory diseases (~100 nM) (Lakey et al., 2016). Further studies are required to unravel threshold concentrations of ROS/EPFR that are harmful to human health. We will add this aspect in the revised manuscript.

References:
Dellinger B., Lomnicki S., Khachatryan L., Maskos Z., Hall R. W., Adounkpe J., McFerrin C., Truong H.: Proceed. Combust. Inst., 31, 521, 2007

Bateman, A. P., Belassein, H., and Martin, S. T.: Impactor Apparatus for the Study of Particle Rebound: Relative Humidity and Capillary Forces, Aerosol Sci. Technol., 48, 42-52, 2014.

Gomes, L., Bergametti, G., Dulac, F., and Ezat, U.: Assessing the actual size distribution of atmospheric aerosols collected with a cascade impactor, J. Aerosol Sci., 21, 47-59, 1990.

Lakey, P. S. J., Berkemeier, T., Tong, H., Arangio, A. M., Lucas, K., Pöschl, U., and Shiraiwa, M.: Chemical exposure-response relationship between air pollutants and reactive oxygen species in the human respiratory tract, Sci. Rep., 6, 32916, 2016.

Seinfeld, J. H., and Pandis, S. N.: Atmospheric chemistry and physics - From air pollution to climate change, John Wiley & Sons, Inc., New York, 2006.

Gehling W., Dellinger B.: Environmentally Persistent Free Radicals and Their Lifetime in PM2.5, Environ. Sci. Tech., 47(15), 8172, 2013

---

## Author Comment (AC3) · 4 Oct 2016

**Response to the comment to the Anonymous Referee #3**

In this study, the authors report size-resolved measurements of EPFR and ROS radicals using EPR spectroscopy and LC-MS for samples collected for one week in Mainz as a demonstration. They show size-dependent variations in the proportion of radicals of ROS and EPFR, with concentrations of both peaking in the accumulation mode, and find that carbon-centered radicals contribute the largest proportions to radical species in ROS for PM1. Using laboratory generated spectra, they further propose that mechanisms for ROS generation in these samples require a combination of transition metals with organic hydropreoxides and quinones. The work is of high technical quality with important implications for understanding atmospheric processes and air quality, and the manuscript is well-written. The work is thus recommended for publication in Atmospheric Chemistry and Physics after the following comments have been addressed.

Response:
We thank the referee for review and very positive evaluation of this manuscript.

General comments regarding the measurement method:
1. Are there any transformation artifacts from the initial measurement of EPFR? For example, if the authors use collocated measurements and analyze ROS directly on the second filter, would they expect to find the same ROS concentrations measured after the filter has been used for quantification of EPFR?

Response:
EPR measurements for the EPFR detection are non-destructive measurements. The microwave radiation used during EPR analysis does not induce any changes in the chemical composition of the sample itself. Indeed, the intensity EPFR signal did not change when monitored over the experimental time. Thus, initial EPFR measurements would not affect subsequent ROS measurements of particle extracts in water.

2. On p.3, line 95, it is stated that "BMPO is an efficient spin-trapping agent [...]." Is the efficiency effectively considered to be 100% for all, or are there biases for certain radicals?

Response:
BMPO is known as a very efficient trapping agent for OH radicals (Tong et al., 2016). In this work, BMPO is assumed to have the same efficiency for all type of radicals. Even though there is no

thorough quantitative data specifically for BMPO trapping efficiency for superoxide and organic radicals, Sueishi et al. (2015) have reported that nitrone-based spin traps have the highest reactivity towards OH and somewhat lower reactivity towards organic radicals and superoxide. Further studies are required to fully address this issue and we will note include the below sentence in the revised manuscript.

3. In the drying process with N2, is it possible that negative artifacts are introduced? How is the dried extract introduced into the EPR spectrometer?

Response:
The text was misleading. The water extracts were exposed to a $N_2$ flow to reduce the volume of the solution to 50 µL. 20 µL are then introduced into a glass capillary for EPR analysis. As $N_2$ is inert towards BMPO-trapped radicals, we do not think artifacts would be introduced. We will clarify it in the revised manuscript.

4. Is the detection limit reported the instrument detection limit, or analytical detection limit derived from blanks?

Response:
The reported detection limit refers to the detection limit of the instrument.

General comments regarding the reported concentrations:
1. Would it be meaningful to plot radiation intensity alongside Fig. 3 to discuss the potential role of photochemistry? For instance, on 02/06/2015, the concentration is also high even though the conditions are presumably cloudy according to descriptions in text. In this regard, the radical concentrations appear to depend on many factors and underscores the benefit of integrating this technique into larger measurement campaigns.

Response:
This is a very good idea as photochemistry may be related to EPFR formation; however, unfortunately we do not have data of radiation intensity.

2. Are the proportions in Fig. 5 meant to be representative of those observed during the entire measurement campaign?

Response:

The proportions in Fig. 5 are referred to samples collected for 48h in June 2015 and extracted in presence of BMPO. We will clarify it in the revised manuscript.

Minor comments:

1. p. 3, line 92: The authors discuss pre-cleaning and weighing, and then discuss particle extraction. It would be helpful if the description were explicit in the pre-sampling and post-sampling procedures.

Response:

Following your suggestion, we will describe explicitly the pre-sampling and the post-sampling procedures as follow:

"Particles were collected on 47 mm diameter Teflon filters (100 nm pore size, Merck Chemicals GmbH). Before sampling, each filter was cleaned and sonicated for 10 min with pure ethanol and ultra-pure water and dried with nitrogen gas before weighing. Teflon filters were weighed four times using a balance (Mettler Toledo XSE105DU) and mounted in the MOUDI. After the sampling, each filter has been conditioned for at least one hour in the lab atmosphere (22 C and 40-50 RH) and weighted four times."

2. p. 5, line 152: This is just a semantic issue, but it would seem more appropriate to say that the values in this work are comparable with the EPFR concentrations measured by Shaltout et al. (2015) - instead of the other way around - since their work preceded this one and sets the precedent to which following studies should be compared.

Response:

Following your comment, we will revise the sentence in the revised manuscript.

3. In the conclusions, the measurement location and period should be restated so the reported concentrations are placed in the proper context.

Response:

We thank referee 3 for pointing this out. We will include this information.

References.

Tong, H., Arangio, A. M., Lakey, P. S. J., Berkemeier, T., Liu, F., Kampf, C. J., Brune, W. H., Pöschl, U., and Shiraiwa, M.: Hydroxyl radicals from secondary organic aerosol decomposition in water, Atmos. Chem. Phys., 16, 1761-1771, 2016.

---

## Author Comment (AC4) · 4 Oct 2016

**Response to the comment to the Anonymous Referee #4**

This paper describes measurements of the concentrations of reactive species embedded in atmospheric aerosol collected from the roof of the MPI. It is a followup to the earlier paper this year by Tong et al which compared field samples to lab samples collected for a shorter period of time and focused on OH generation. The measurement method consists of extracting soluble molecules from the particles and reacting them with a scavenger. This is essentially a physical chemistry paper and my comments are from that perspective. This paper will be publishable after some edits to respond to the following comments.

Response:
We thank the referee for review and positive evaluation of this manuscript.

I found the terminology used by the authors to be confusing in places. Through use of words such as "we have also characterized and quantified ROS including OH, superoxide (O2-) and carbon- and oxygen-centered organic radicals, which were released upon extraction of the particle samples in water." the reader could conclude that the radicals are persistently present in the particle, rather than being formed by reaction of water with precursors during the extraction process and later scavenged. The multiple chemical steps involved in the experiment lend an ambiguity to how to relate the lab processes that are responsible for formation of detectable spins to atmospheric and physiological processes. As another example, on pages 4-5 the authors talk about spins per microgram but do not define this quantity. I presume they mean spins detected by extraction from a sample of this mass using the protocol described. The concentration of the scavenger is not given so it is not clear how closely this process is controlled or how repeatable it is. Clarification of the terminology and relation of the experimental conditions to those found in the lung and in clouds, for example, would be helpful.

Response:
Following your suggestion, we will make it clear that we measured ROS formed upon extraction into water throughout the revised manuscript. The unit for EPFR and ROS concentrations used in this study is spins $\mu g^{-1}$, which indicates the number of spins (or radicals) produced per unit of particle mass. We will add the following sentence in the revised manuscript.
"Concentrations of EPFR and ROS are reported in the unit of spins $\mu g^{-1}$, which indicates the number of spins (or radicals) per $\mu g$ of particle mass."

The concentration of the scavenger was specified in the method section (350 μL of 20 mM BMPO was used). The experimental procedure was controlled well and repeatable (e.g., Tong et al., 2016).

(2) Only particles smaller than 1 micron contain extractable ROS material. Do the authors understand why this is? Since peroxides are photo labile I might have expected the opposite - the larger, more optically opaque particles would have more precursors than the smaller ones assuming the extraction processes work the same way for all particle sizes. Some discussion of the size effects would be useful.

Response:

ROS concentrations are indeed smaller for particles in the coarse mode, but ROS were formed also by particles larger than 1 μm, as shown by the red line in Fig. 2, Fig. 4a and Fig. 5. The size-dependence was discussed in L266 – 273 ("SOA particles, which may contain large amounts of organic hydroperoxides, account for a major fraction in PM1 (Jimenez et al., 2009). SOA compounds may also coat coarse particles such as biological particles (Pöhlker et al., 2012). As shown in Fig. 2, semiquinones are mostly contained in submicron particles but not in coarse particles. Thus, the release of a variety of ROS species are most likely due to the interactions of organic hydroperoxides, semiquinones, and transition metal ions, whereas the dominance of OH radicals in coarse particles may be due to the decomposition of organic hydroperoxides in the absence of semiquinones. ").

(3) The reactive oxygen species released and scavenged during the analytical protocol are well known to have rich chemistry in water and very different reactivities compared to each other. Have the authors determined how efficiently are they being detected (absolute and relative values)?

Response:

We are aware that ROS chemistry is quite complex. We are currently making efforts on synthesizing possible reactions involving ROS, organic hydroperoxides, quinones and transition metals and developing the kinetic model. We intend to present these results in the follow-up study. Regarding the detection efficiency of ROS, BMPO is known as a very efficient trapping agent for OH radicals (Tong et al., 2016). In this work, BMPO is assumed to have the same efficiency for all type of radicals. Even though there is no thorough quantitative data specifically for BMPO trapping efficiency for superoxide and organic radicals, Sueishi et al. (2015) have reported that nitrone-based spin-traps have the highest reactivity towards OH followed by carbon-centered radicals, oxygencentered organic radicals, and superoxide. Further studies are required to fully address this issue and we will note include the below sentence in the revised manuscript.

(4) On page 9 line 269ff there is a section discussing implications for aerosol chemistry and lung chemistry. Since there are no data in this paper specifically looking at these implications but there are in Tong I recommend the reader referred back to the earlier paper instead.

Response:

Yes, we refer this aspect to Tong et al., (2016). This study itself did not show that ROS can be generated in the lung lining fluid containing antioxidants, but it did show that the particles can form ROS in water. Several previous studies have shown that redox-active components such as transition metals and quinones can induce formation of ROS species upon interactions with lung antioxidants (Charrier et al., 2014; Charrier and Anastasio, 2011). We will clarify it in the revised manuscript as below:

"Previous studies have shown that redox-active components such as transition metals and quinones can induce ROS formation in surrogate lung lining fluid upon interactions with antioxidants (Charrier et al., 2014; Charrier and Anastasio, 2011). This study also implies that ROS may be released in lung lining fluid upon inhalation and respiratory deposition of atmospheric aerosol particles."

**References.**

Tong, H., Arangio, A. M., Lakey, P. S. J., Berkemeier, T., Liu, F., Kampf, C. J., Brune, W. H., Pöschl, U., and Shiraiwa, M.: Hydroxyl radicals from secondary organic aerosol decomposition in water, Atmos. Chem. Phys., 16, 1761-1771, 2016.